# OpenReview forum: "House of Cards: Massive Weights in LLMs"
_ICLR.cc/2025/Conference — Submitted to ICLR 2025_

### Official Review · Reviewer_RoCp · 2024-11-03

**Soundness:** 1
**Presentation:** 3
**Contribution:** 2
**Rating:** 3
**Confidence:** 5

**Summary:**

The paper investigates the phenomenon of “massive activations” in large language models (LLMs), identifying specific “massive weights” in the intermediate layers that significantly impact model performance. The authors show that these massive weights, although few in number, are highly influential during the pre-training process. To leverage this insight, the paper introduces a method called “Massive Weights Curriculum Dropout” (MacDrop), which applies dropout specifically to these massive weights during fine-tuning. By starting with a high dropout probability and gradually reducing it, MacDrop reduces the model’s reliance on massive weights, leading to improved performance on zero-shot downstream tasks and modest improvements in generation tasks. Through extensive experiments across different models and tasks, the paper demonstrates MacDrop’s effectiveness as a simple, parameter-efficient fine-tuning technique that can be easily integrated into existing workflows.

**Strengths:**

1. Practical and Simple Method: MacDrop is presented as a straightforward, plug-and-play approach that can be applied with minimal modifications to fine-tuning pipelines.
2. High-Quality Writing and Visuals: The paper is well-written and supported by clear figures, making the complex concepts and experimental findings accessible and visually engaging.
3. Clear Experimental Results: The paper provides comprehensive experimental results across various LLM architectures and tasks.

**Weaknesses:**

1.  Insufficient Comparative Analysis with Other Techniques
The paper does not provide sufficient comparisons between MacDrop and other dropout or regularization techniques, such as Deja Vu[1] and N:M Sparsity[2]. Without such comparisons, it is difficult to assess whether MacDrop offers any distinct advantages over existing approaches. Including these baselines would make the findings more robust and allow for a clearer understanding of MacDrop’s relative performance.
2. Lack of Theoretical Justification for Massive Weights' Impact.
The paper identifies the phenomenon of massive weights and their importance, but it lacks a strong theoretical explanation as to why these weights have such a significant influence on the model’s performance. Establishing a deeper theoretical foundation could enhance the credibility and understanding of why massive weights dominate model behavior.
3. Unclear Robustness Improvement Claims
Although MacDrop aims to reduce dependence on massive weights, the paper lacks an analysis of whether this actually leads to improved robustness. For example, evaluating the model’s performance under adversarial attacks or noisy inputs would provide evidence of whether MacDrop indeed enhances the robustness of the model. This is crucial to establish the value of MacDrop beyond improving performance in specific tasks.
4. Lack of Practical Implications for Deployment
The paper does not adequately discuss the practical implications of the identified massive weights and the MacDrop method for real-world deployment. For instance, how does the presence of massive weights affect the robustness of models in production settings? Are there potential risks for using such models? Addressing these concerns would help bridge the gap between theory and practical application.
5. Discrepancy between Perplexity and Auto-eval Scores: In Table 1, the paper reports a significant loss in perplexity with MacDrop, while auto-evaluation scores show minimal loss or slight improvement on certain metrics, which is unusual and warrants further investigation to understand the divergence in these evaluation metrics. For example, based on Llama-3-8B, the ppl of "top-5 retaining" is as large as 20.95, with which the model can't even generate a logical diaglogue  in my experimental experimence, while its average score is 71.0 (only drop 2.0), is it possible?
6. Limited Generalizability of BOS Token Magnitudes: Figure 2 shows that the magnitudes associated with the BOS token do not generalize to other tokens, which contradicts the underlying assumption of MacDrop, potentially limiting its effectiveness across different tokens and contexts.

reference:
[1] Deja Vu: Contextual Sparsity for Efficient LLMs at Inference Time
[2] Progressive Gradient Flow for Robust N:M Sparsity Training in Transformers

**Questions:**

1. Discrepancy between Perplexity and Auto-evaluation Scores: There is a significant loss in perplexity with MacDrop, while auto-evaluation metrics show only minimal losses or slight improvements for some tasks. Could the authors provide further analysis on why these two metrics diverge so drastically? Additional clarification on this point could help to interpret the overall impact of MacDrop on model performance.
2. Generalizability of BOS Token Findings: As shown in Figure 2, the magnitudes associated with the BOS token do not appear to generalize to other tokens. This observation seems to contradict the underlying principle of MacDrop, which relies on massive weights associated with the BOS token to generalize across the model. Can the authors explain or justify how MacDrop remains effective given this discrepancy? Further explanation could strengthen the validity of the method across diverse contexts.
3. Using in Long Context: Specifically, when adapted to long context area, the tokens near EOS tokens usually has differential feature distrubutions compared with BOS tokens, is this case, will MacDrop still work?
4. Reproduction: Will the authors make the code for MacDrop publicly available? I would appreciate it if the code could be made accessible before the next review round.

---

> ### Author Response · Authors · 2024-11-22
>
> We are grateful for the valuable and constructive feedback. We have uploaded a revised manuscript, where the modified parts are red-colored fonts, and organized our responses corresponding Weaknesses (W) and Questions (Q).
> ### W1) Insufficient Comparative Analysis with Other Techniques
> - We thought that the appropriate algorithms for our comparison are those that do not freeze pre-trained weights during PEFT, similar to MacDrop.
> - From this perspective, the papers you recommended do not seem to align with our algorithm. They focuse primarily on efficient inference using sparsity and do not use PEFT.
> - However, our ablation study in **Section 4.3** provides results for different dropout methods based on the scope of dropout and curriculum approaches. We think these variations can serve as a sufficient baseline.
>
> ### W2) Lack of Theoretical Justification for Massive Weights' Impact
> - We fully understand the value of adding theoretical justification, as suggested.
> - However, conducting such a theoretical analysis presents significant challenges. For example, the impact of massive weights may remain consistent with model scaling in some cases (e.g., the LLaMA-3 family), but it can also vary in other cases (e.g., the Phi-3 family), shown in **Figure 4**.
> - Nonetheless, we have made every effort to ensure the validity of our observations through extensive experiments and verification across various settings using diverse and up-to-date range of models.
>
> ### W3) Unclear Robustness Improvement Claims
> - Thank you for the valuable feedback. As you mentioned, MacDrop aims to reduce dependence on massive weights during PEFT. To verify whether the dependency on massive weights is reduced, the top-k zeroing attack in **Section 2.3** can be used.
> - Specifically, we examine the performance changes when applying the top-3 zeroing attack to Llama-3-8B and Mistral-7B with and without MacDrop, and have updated the results in **Appendix G**.
> - The models with MacDrop exhibit significantly better performance under attack, indicating better robustness, as we expected.
>
> ### W4) Lack of Practical Implications for Deployment
> - We appreciate your feedback that helps enhance the insights of our research.
> - Our research demonstrates, through various open-sourced LLMs, that in many cases, models can be easily disrupted by zeroing massive weights **if access to their weights is available** (**Section 2.3**). Subsequently, we improve robustness against such attacks through MacDrop (**Appendix G**).
> - We think the existence of massive weights does not inherently present a risk if the model weights remain undisclosed. However, we believe that model developers should understand these internal mechanisms and take responsibility for ensuring robustness. In this vein, an approach similar to MacDrop is used to improve robustness.
>
> ### W5 & Q1) Discrepancy between Perplexity and lm-eval Score
> - First of all, we clarify that Table 1 does not represent the performance degradation caused by MacDrop but rather the performance degradation caused by the two types of attacks. Please refer to the caption of **Table 1**.
> - As you mentioned, if the perplexity is around 20, the model is unable to provide a logical dialogue.
> - Understanding the evaluation method of lm-eval seems crucial to solving this issue. lm-eval evaluates **by calculating the loglikelihood for provided options and selecting the one with the highest loglikelihood among them** for scoring. In analogy, it is a situation where the problem would have been unsolvable if it were a subjective question but becomes solvable because it is multiple-choice.
> - Similarly, as observed in Tables 1 and 2 of [1], for per-tensor dyanmic quantization of LLaMA-3-8B, the perplexity increases from 6.13 to 23.86, while the zero-shot accuracy decreases from 68.83 to 58.94.
>
> ### W6 & Q2) Limited Generalizability of BOS Token Magnitudes
> - What we are emphasizing is not that the phenomenon occurring in the BOS token appears in other tokens as well, but that the phenomenon observed in the BOS token is consistent across different models.
> - During rebuttal, we have clarified **Lines 183-200** about generalzability of the BOS token placed at the starting position, and added all of the results across various LLMs on **Appendix C**.
>
> ### Q3) Using in Long Context
> - Thank you for your valuable suggestion. We have added the performance on LongBench.
> - The improvements brought by MacDrop are more pronounced on long context tasks, which increases the average score across 16 tasks by up to approximately 1.5 points. Please refer to **Appendix H**.
>
> ### Q4) Reproduction
> - We are preparing to release the code, but due to company regulations, it may be difficult to disclose within the discussion period.
>
> ---
> [1] Prefixing Attention Sinks can Mitigate Activation Outliers for Large Language Model Quantization, EMNLP 2024

---

> > ### Comment · Reviewer_RoCp · 2024-11-26
> > **Concerns Regarding Table 1 Results and Methodology**
> >
> > Thank you for your response. I have been working in the field of large language modeling for over three years, and I have significant concerns about the results presented in Table 1 and their inherent reasonability. My concerns can be summarized as follows:
> > 1. Experimental Results in Table 1:
> > - As mentioned in the earlier question, the perplexity improvements when using the top-5 retaining strategy are substantial. However, these improvements seem contradictory to the fact that models employing this strategy are empirically has terrible performance and even unable to generate coherent dialogue.
> > - While downstream tasks are evaluated by calculating the log-likelihood of the provided options and selecting the one with the highest log-likelihood, it raises a critical question: how can models with impaired dialogue coherence (using top-5 retaining) demonstrate comparable performance to the original models, whose perplexity is significantly better?
> > - Please verify the implementation of lm-eval to ensure whether the retaining strategy is applied consistently across both the generation process and the pre-filling phase. If the retaining strategy is only applied during the generation process and not during pre-filling, the lm-eval scores in Table 1 would solely reflect the performance of pre-filling, without considering the generation aspect. This discrepancy could lead to misleading results, as the scores in lm-eval may not accurately represent the model’s true performance. If this is the case, the experimental results reported in this paper based on lm-eval would be invalid and require careful reassessment.
> >
> > 2.	Reasonability of the MacDrop Method
> > - The MacDrop method seems useful only if the pattern observed in the BOS token applies consistently to other tokens as well. As acknowledged by the authors in Lines 078-082, if the pattern does not hold for other tokens, the computation of weights for those tokens would require retaining all information. However, in the rebuttal phase, the authors stated:
> > “What we are emphasizing is not that the phenomenon occurring in the BOS token appears in other tokens as well, but that the phenomenon observed in the BOS token is consistent across different models.” This statement is ambiguous and does not clearly address the applicability of the BOS token pattern to other tokens.
> >
> > I hope these points contribute to refining the discussion and ensuring the validity of the results presented. However, if these questions remain unresolved, I am inclined to adjust my Confidence score to 5.

---

> ### Author Response · Authors · 2024-11-26
> **Replying to Concerns Regarding Table 1 Results and Methodology**
>
> ### 1) Experimental Results in Table 1
> - For zero-shot accuracy using lm-eval, **only the pre-filling phase** is employed to compute log-likelihood.
> - This is the example from the ARC-easy.
>   - context : "Question: Which statement best explains why photosynthesis is the foundation of most food webs?\nAnswer:"
>   - continuation
>     - choice 1 : " Sunlight is the source of energy for nearly all ecosystems."
>     - choice 2 : " Most ecosystems are found on land instead of in water."
>     - choice 3 : " Carbon dioxide is more available than other gases."
>     - choice 4 : " The producers in all ecosystems are plants."
>   - answer : choice 1
> - Then, we can make four sentences:
>   - sentence 1 : "Question: Which statement best explains why photosynthesis is the foundation of most food webs?\nAnswer: Sunlight is the source of energy for nearly all ecosystems."
>   - sentence 2 : "Question: Which statement best explains why photosynthesis is the foundation of most food webs?\nAnswer: Most ecosystems are found on land instead of in water."
>   - sentence 3 : "Question: Which statement best explains why photosynthesis is the foundation of most food webs?\nAnswer: Carbon dioxide is more available than other gases."
>   - sentence 4 : "Question: Which statement best explains why photosynthesis is the foundation of most food webs?\nAnswer: The producers in all ecosystems are plants."
> - Finally, the loglikelihood for **continuation** is calculated without auto-regressive decoding phase.
>   - For Llama-3-8B, [-18.75855827331543, -30.829565048217773, -30.68844223022461, -20.0650691986084] → "choice 1" (right)
>   - For Llama-3-8B (top-5 retaining), [-17.360136032104492, -27.842376708984375, -27.37234878540039, -20.682714462280273] → "choice 1" (right)
>   - For Llama-3-8B (top-5 zeroing), [-60.779014587402344, -58.72906494140625, -53.693702697753906, -52.90177917480469] → "choice 4" (wrong)
>
> - Furthermore, **the last example of Figure 1(B)** demonstrates that Llama-3-8B-Instrcut (top-5 retaining) generates plausible sentences. We provide the code.
> ```
> import torch
> from transformers import AutoTokenizer, AutoModelForCausalLM, AutoConfig, AutoProcessor
>
> model_path = "meta-llama/Meta-Llama-3-8B-Instruct"
>
> access_token = "hf_XXXXX"
> config = AutoConfig.from_pretrained(model_path, trust_remote_code=True, token=access_token)
> tokenizer = AutoTokenizer.from_pretrained(model_path, trust_remote_code=True, token=access_token)
> processor = AutoProcessor.from_pretrained(model_path, trust_remote_code=True, token=access_token)
> model = AutoModelForCausalLM.from_pretrained(model_path, torch_dtype=config.torch_dtype, device_map="cpu", trust_remote_code=True, token=access_token)
>
> model.eval()
> model.cuda()
>
> massive_indices = torch.tensor([2427, 198, 6412, 12657, 591]).to(model.device) # for top-5 zeroing
> nonmassive_indices = torch.arange(config.intermediate_size).to(model.device)
> nonmassive_indices = nonmassive_indices[~torch.isin(nonmassive_indices, massive_indices)] # for top-5 retaining
>
> model.model.layers[1].mlp.up_proj.weight.data[massive_indices, :] *= 0 # or, nonmassive_indices
> model.model.layers[1].mlp.gate_proj.weight.data[massive_indices, :] *= 0 # or, nonmassive_indices
>
> prompt = [
>     {"role": "user", "content": "Describe a vivid and unique character, using strong imagery and creative language."},
> ]
> prompt = tokenizer.apply_chat_template(prompt, tokenize=False)
> inputs = processor(prompt, return_tensors="pt")
>
> input_ids = inputs.input_ids.cuda(model.device)
> attention_mask = inputs.attention_mask.cuda(model.device)
>
> outputs = model.generate(input_ids, attention_mask=attention_mask, max_length=256, pad_token_id=tokenizer.eos_token_id)
> print (tokenizer.batch_decode(outputs, skip_special_tokens=True)[0])
> ```
> - Additionally, we already provided a proper reference, where the similar results about perplexity and zero-shot accuracy appear.
>
>
> ### 2) Reasonability of the MacDrop Method
> - In fact, we do not understand the "underlying principle of MacDrop" that you are referring to.
> - The underlying principle of MacDrop that we assert is simply that **massive weights have been dominantly learned during pre-training**.
> - Prior research has revealed that massive activations occur in specific feature dimensions of **certain tokens or position**.
>   - Therefore, **we have never expected all tokens to exhibit the same magnitude distribution**; this is a fact that has been revealed in prior research.
> - Rather, we further clarify the previous findings by stating that **the BOS token placed at the first position** has always massive activations **across diverse LLMs**. (Lines 183-200)
>   - Namely, we intend the generality of the BOS token exhibiting massive activations across various LLMs, rather than the generality between the BOS token and other tokens.
> - Therefore, we proposed an algorithm based on this token because we know that it has a significant impact on LLMs, rather than attempting to achieve the same effect for all tokens.

---

> ### Comment · Area_Chair_45mk · 2024-11-30
>
> Dear Reviewer,
>
> Could you kindly respond and indicate whether authors have addressed your concerns?
>
> Thanks, AC

---

> > ### Comment · Reviewer_RoCp · 2024-11-30
> >
> > First, I would like to thank the authors for their time and effort in addressing my questions. As an LLM researcher, I am thoroughly familiar with the code and strategies employed in the evaluation tasks discussed in this paper and have independently implemented the evaluation pipeline for my organization.
> >
> > However, I remain skeptical about the evaluation scores presented in the paper, particularly due to the significant perplexity drop observed with MacDrop. While the method aims to reduce computation in the FFN of **a single layer** (as detailed in Appendix B and the provided code, where the computational gains are **less than** 1/32 = 3.13% for Llama-3-8B, 1/80 = 1.25% for Llama-3-70B, and 1/126 = 0.79% for Llama-3.1-405B), the extent of the perplexity drop appears substantial and raises concerns about its impact.
> >
> > Based on these observations, I have decided to maintain my original rating.

---

> ### Author Response · Authors · 2024-12-01
> **Response to Reviewer RoCp’s Final Assessment**
>
> The reviewer RoCp has made the final decision. We aim to demonstrate that this reviewer has **completely misunderstood** our paper.
>
> - The reviewer refers to "the significant perplexity drop observed with MacDrop" as the rationale for their evaluation.
>   - However, our paper does not provide any perplexity results for MacDrop anywhere.
>   - To clarify this, **in our first response**, we explicitly mentioned "First of all, we clarify that Table 1 does **not represent the performance degradation caused by MacDrop** but rather the performance degradation caused by the two types of attacks. Please refer to the caption of Table 1." This indicates that the reviewer did not properly read our response.
>   - Moreover, while they raised a question regarding the issue between perplexity and zero-shot accuracy, their response to our reply merely states that they are familiar with the evaluation pipeline as an LLM researcher. This leaves us completely unclear about what they intended to convey in their review.
> - Furthermore, the reviewer RoCp states that "MacDrop aims to reduce computation."
>   - Regarding this, we make it clear that **we have never mentioned that MacDrop is intended to reduce computation**.
>     - This sentence helps us understand why the reviewer RoCp suggested entirely unrelated references in their initial review.
>   - Rather, we emphasized that **even a very small part of the LLM can have a significant impact**.
>   - Additionally, a review with the reviewer ChSN reveals that MacDrop is an algorithm that introduces very minimal overhead.
> - Finally, it was confirmed that the reviewer raised their confidence score from 4 to 5.
>   - The reviewer RoCp stated in their second reaction that they would raise their confidence score if the issue remained unresolved. They are treating their confidence score as if it were their weapon in this statement.
>   - We do not understand the connection between resolving the issue and their confidence score.
>   - We hope that they fulfill their role as a reviewer with greater kindness and deeper understanding.
> ---
>
> Dear AC/SAC/PCs.
> We kindly ask for this to be taken into consideration when meta-reviewing.
>
> Best regards,
> Authors.

---

### Official Review · Reviewer_N9MC · 2024-11-03

**Soundness:** 2
**Presentation:** 3
**Contribution:** 2
**Rating:** 5
**Confidence:** 4

**Summary:**

This paper examines the massive weights in LLMs and investigates their significance. Through targeted interventions, the authors demonstrate that these weights, while constituting only a small fraction of the total, are crucial for model performance. Building on these findings, they introduce a parameter-efficient algorithm for fine-tuning LLMs.

**Strengths:**

The experiments in Section 2 are very insightful. The paper's motivation is based on the empirical analysis massive activations, and then shifts to the relation between massive activations to massive weights, making the motivation very good.

**Weaknesses:**

1. The analysis of the influence of massive weights is very empirical, limiting the paper's contribution and novelty. To enhance rigor and generalizability, the authors should include a more solid mathematical analysis to better elucidate the roles of massive weights.

2. In Tables 2 and 3, the authors compare their approach with only a single vanilla LoRA/DoRA baseline, which appears insufficiently comprehensive.

3. In Table 3, the performance improvements with "w/ MacDrop" are marginal compared to the LoRA/DoRA baseline.

**Questions:**

Please see the Weaknesses.

---

> ### Author Response · Authors · 2024-11-22
>
> We are grateful for the valuable and constructive feedback. We have uploaded a revised manuscript, where the modified parts are red-colored fonts, and organized our responses corresponding Weaknesses (W).
> ### W1) Theoretical analysis on massive weights
> - We fully understand the value of adding theoretical justification, as suggested.
> - However, conducting such a theoretical analysis presents significant challenges. For example, the impact of massive weights may remain consistent with model scaling in some cases (e.g., the LLaMA-3 family), but it can also vary in other cases (e.g., the Phi-3 family), shown in **Figure 4**.
> - Nonetheless, we have made every effort to ensure the validity of our observations through extensive experiments and verification across various settings using diverse and up-to-date range of models.
>
> ### W2) Lack of baselines
> - MacDrop is a method that can be applied orthogonally to PEFT that freezes pre-trained weights (e.g., LoRA/DoRA). Therefore, through Tables 2 and 3, we presented the results comparing the cases when MacDrop is applied to LoRA/DoRA and when it is not.
> - What we aimed to demonstrate is not that MacDrop is better than LoRA/DoRA, but that LoRA/DoRA performs better when MacDrop is applied compared to when it is not.
> - In this vein, the algorithms to be compared with MacDrop should be approaches that modify the pre-trained network during PEFT. Therefore, the variations in scope or curriculum of the dropouts used in the ablation study of **Section 4.3** can serve as a sufficient baseline.
>
> ### W3) MacDrop's inability on generation tasks.
> - The aspect you pointed out may be considered a limitation of MacDrop.
> - Nevertheless, during the rebuttal phase, we additionally evaluate our method on different tasks, to demonstrate the superiority of MacDrop.
> - The improvements brought by MacDrop are more pronounced on long context tasks, which increases the average score across 16 tasks by up to approximately 1.5 points. Please refer to **Appendix H**.
> - Furthermore, we have examined the robustness of the trained models through top-$k$ zeroing attack and have added the results in **Appendix G**. The models with MacDrop exhibit significantly better performance under attack, indicating reduced dependence on massive weights.

---

> ### Comment · Reviewer_N9MC · 2024-11-26
>
> Thank you for you clarification. I appreciate the additional experiments in the Appendix. However, this work lacks rigorous mathematical analysis, and I deem it as a major weakness. Therefore, I maintain the score.

---

### Official Review · Reviewer_ChSN · 2024-11-04

**Soundness:** 3
**Presentation:** 4
**Contribution:** 3
**Rating:** 8
**Confidence:** 4

**Summary:**

This paper analyzes the phenomenon of massive weights in LLMs. These weights are related to previously-observed massive activations. The authors find that massive weights are critical for model performance. They introduce a novel dropout method, called MacDrop, which reduces reliance on massive weights and improves performance.

**Strengths:**

- This paper includes an in-depth analysis of massive weights in LLMs and their relationship to massive activations.
- The investigation builds upon previous observations around massive activations and attention sink tokens. Rather than previous work which focused on token location, this work explores how the bos token, no matter its location in the sentence, emerges as an attention sink.
- The authors present a set of ablation studies demonstrating the effects of preserving the top k massive weights and removing the rest, or dropping the top k massive weights and keeping the remaining weights. They compare these results to baselines on several relevant tasks.
- The MacDrop method performs dropout to weights on a curriculum. The aim of this method is to reduce the dependence on massive weights and therefore achieve higher performance on downstream tasks. These ideas are validated through experimental results.

**Weaknesses:**

- MacDrop’s inability to improve performance on generation tasks.
- There does not seem to be any analysis validating the hypothesis that there are fewer massive weights when training with MacDrop. Some demonstration of this would be nice.

**Questions:**

- What is the additional computational cost of performing MacDrop?

---

> ### Author Response · Authors · 2024-11-22
>
> We are grateful for the valuable and constructive feedback. We have uploaded a revised manuscript, where the modified parts are red-colored fonts, and organized our responses corresponding Weaknesses (W) and Questions (Q).
>
> ### W1) MacDrop's inability on generation tasks.
> - Thank you for the valuable feedback. In fact, the aspect you pointed out may be considered a limitation of MacDrop.
> - Nevertheless, during the rebuttal phase, we additionally evaluate our method on different tasks, to demonstrate the superiority of MacDrop.
> - The improvements brought by MacDrop are more pronounced on long context tasks, which increases the average score across 16 tasks by up to approximately 1.5 points. Please refer to **Appendix H**.
>
> ### W2) Fewer massive weights when training with MacDrop
> - We defined the rows in the projection matrix $W_{up}$ (and $W_{gate}$, if it exists) that correspond to the indices of the top-k magnitudes in the intermediate state as top-k massive weights, thus the number of them is $k \times d$, where $d$ is the dimensions of hidden state. Please refer to **Lines 233-236**.
> - When training with MacDrop, the activated number of massive weights is $k \times d \times (1-p)$, where $p$ is dropout probability. Therefore, there are fewer massive weights when training with MacDrop, to reduce dependence on massive weights during PEFT.
> - To observe the effects of using fewer activated massive weights during fine-tuning, we have examined the robustness of the trained models through top-$k$ zeroing attack and have added the results in **Appendix G**.
> - The models with MacDrop exhibit significantly better performance under attack, indicating reduced dependence on massive weights.
>
> ### Q1) MacDrop's additional computational cost
> - The additional computational cost of performing MacDrop is slight.
> - In **Algorithm 1**, Lines 5, 6, 7, 9 are the additional computation for MacDrop, but Line 8 originally exists for PEFT. When we implement Lines 1-3 in practice, we precomputed the massive indices and loaded them. The time required for this process is minimal (less than one minute) because it only involves the BOS token.
> - According to the training log, for Llama-3-8B using LoRA, fine-tuning of 579 steps takes 48 minutes and 31 seconds with the original setup, while it takes 51 minutes and 55 seconds when MacDrop is applied on 8xA100 GPUs. There is approximately an additional computational cost of **0.35 seconds per step**.

---

> > ### Comment · Reviewer_ChSN · 2024-11-26
> > **Thank you**
> >
> > Thank you to the authors for addressing my concerns. I will maintain my previous score.

---

### Official Review · Reviewer_6SXM · 2024-11-04

**Soundness:** 2
**Presentation:** 3
**Contribution:** 2
**Rating:** 5
**Confidence:** 4

**Summary:**

This paper reveals that large language models (LLMs) have a bias towards specific tokens due to massive activations in their hidden states, which stem from the intermediate state of early layer feed-forward network modules. The authors define top-k massive weights as those with the greatest impact on these activations and propose MacDrop, a method that employs dropout on these weights during fine-tuning to reduce their influence. By starting with a high dropout rate and gradually decreasing it, MacDrop improves LLM performance on zero-shot tasks and generations, offering a parameter-efficient fine-tuning strategy that enhances model robustness.

**Strengths:**

1. The paper identifies a new and critical phenomenon in LLMs where specific feature dimensions in hidden states exhibit massive activations, leading to an overemphasis on certain tokens. This discovery contributes to a deeper understanding of potential biases within LLMs.
2. The authors provide detailed experiments and analysis on massive activations and massive weights, in various LLMs.
3. MacDrop is introduced as a plug-and-play method that applies curriculum dropout to pre-trained massive weights during fine-tuning, which is an innovative approach to reducing reliance on these dominant weights.

**Weaknesses:**

1. Novelty.

1.1 In this paper, the massive weights are rows in weight matrics related to the massive activations. It is similar to Wanda-sp proposed in [1], which is modified from [2]. From this perspective, the novelty of this paper is limited.

1.2 Moreover, the proposed fine-tuning method MacDrop is similar to LoRA. The only difference is that MacDrop uses a curriculum strategy to gradually reduce the dropout rate of the massive weights. However, the improvement of MacDrop over LoRA is less significant.

2. As the authors discovered, different models have different robustness to the massive weights and the phenomenon of massive weights is not as universal as massive activation [3]. It might also imply that MacDrop only works on models that are sensitive to massive weights, e.g. Llama-3-8B and Mistral-7B.

3. The experiments are not sufficient, for example,

3.1. Does the choice of the dataset have any influence on the massive activations and massive weights?

3.2. This paper illustrates the Llama-3-8B model shows the phenomenon that the bos token always has massive activations, regardless of its position. Do other models share the same phenomenon?


[1] Fluctuation-based Adaptive Structured Pruning for Large Language Models. AAAI 2024

[2] A simple and effective pruning approach for large language models. ICML 2023

[3] Massive activations in large language models. ICML 2024

**Questions:**

Please refer to those in the Weakness section.

---

> ### Author Response · Authors · 2024-11-22
>
> We are grateful for the valuable and constructive feedback. We have uploaded a revised manuscript, where the modified parts are red-colored fonts, and organized our responses corresponding Weaknesses (W).
>
> ### W1.1) Similarity between massive weights and Wanda-SP (structured pruning).
>   - Thank you for recommending related papers. As you mentioned, there is a similarity in that the massive weights are defined in an activation-aware manner.
>   - Nonetheless, we would like to emphasize that the massive weights we defined are characterized within a **specific single** layer, unlike Wanda-SP [1,2] or AWQ [3], which identify important weights within every linear layer. The massive weights would be included among the weights chosen through methods such as Wanda or AWQ.
>   - To clarify this point, we have added relevant information in **Lines 505-510 of Section 5**.
>
> ### W1.2) Similarity between MacDrop and LoRA.
>   - We do not consider MacDrop to be similar to LoRA/DoRA; rather, MacDrop is a method that can be applied orthogonally. Please refer to **Lines 345-346**.
>   - Therefore, we presented the results comparing the cases when MacDrop is applied to LoRA/DoRA and when it is not.
>   - In Tables 2 and 3, what we aimed to demonstrate is not that MacDrop is better than LoRA/DoRA, but that LoRA/DoRA performs better when MacDrop is applied compared to when it is not.
>   - Furthermore, during the rebuttal phase, we evaluate our method on long context tasks, where the improvements brought by MacDrop are more pronounced. It increases the average score across 16 tasks by up to approximately 1.5 points. Please refer to **Appendix H**.
>
> ### W2) MacDrop on LLMs that are not sensitive to massive weights.
>   - During the rebuttal period, we conduct zero-shot downstream experiments on various LLMs with MacDrop and have attached the results on **Appendix F**.
>   - As you mentioned, the results show that MacDrop is not effective for models that are not sensitive to massive weights (e.g., phi-3-medium / Gemma-2 family).
>   - In fact, we considered this aspect to be a potential limitation, as stated in **Lines 334-335**. We are thankful for your insightful suggestion, which has brought clarity to this limitation.
>   - We are currently evaluating the Gemma models and will provide an update within three days. (updated.)
>
> ### W3.1) Influence of dataset on massive weights and massive activations.
>   - The phenomena related to "massive" have been identified as occurring during the pre-training process, in [4]. However, we were unable to verify the pre-training process due to constraints on resources.
>   - Nevertheless, we observe that massive activations appear across Llama-2/3, Mistral/Mixtral, Phi-3 families and we recognize that these models were trained on different pre-training datasets. This implies that while the dataset composition influences model behavior, the architectural choices and training dynamics might play significant roles related to the "massive" phenomena.
>   - Therefore, we aimed to analyze, from an architectural perspective (e.g., dropout and layer normalziation), why the massive-related phenomena appear weakly in Phi-3-medium and Gemma-2 models. Please refer to **Lines 316-333**.
>
> ### W3.2) Phenomenon by the BOS token across different models.
>   - Thank you for the valuable observations. Upon examining various LLMs, we clarify that massive activations occur **when the BOS token is placed at the first position**.
>   - In [4], it has been stated that massive activations always occur at the starting position; however, there are model families where this is not the case, such as Mistral/Mixtral and Gemma-2 families.
>   - In summary, among Llama-2, Llama-3, Mistral/Mixtral, Phi-3, Gemma-2 families:
>     - Llama-2, Llama-3, and Phi-3 families have massive activations at the first position.
>     - Llama-3, Mistral/Mixtral families, and Gemma-2-2/9b models have massive activations at the BOS token.
>     - All families have massive activations at the BOS token placed at the first poistion.
>   - We have clarified **Lines 183-200** following these results, and added all of the results across various LLMs on **Appendix C**.
>   - Nevertheless, **please note that since we consistently utilized only the BOS token, which has a length of 1, there will be no changes to the results of our analysis**.
>
> ---
> [1] A simple and effective pruning approach for large language models. ICML 2023
> [2] Fluctuation-based Adaptive Structured Pruning for Large Language Models. AAAI 2024
> [3] Activation-aware Weight Quantization for LLM Compression and Acceleration, MLSys 2024
> [4] Massive Activations in Large Language Models, COLM 2024

---

> > ### Comment · Area_Chair_45mk · 2024-11-30
> >
> > Dear Reviewer,
> >
> > Could you kindly respond and indicate whether authors have addressed your concerns?
> >
> > Thanks, AC

---

> ### Author Response · Authors · 2024-12-01
>
> Dear Reviewer 6SXM,
>
> Thank you once again for taking the time and effort to review our work. As the deadline for the discussion period is approaching, we would greatly appreciate it if you could review our responses.
>
> Thank you!
>
> Best regards, Authors

---

> > ### Comment · Reviewer_6SXM · 2024-12-02
> >
> > Dear Authors,
> >
> > Thank you for your clarification. I appreciate the efforts the authors have made during the discussion period. However, I still have the following concerns:
> >
> > - The improvement of MacDrop over LoRA and DoRA appears to be less significant, which might diminish the perceived contribution of this work.
> > - MacDrop seems to be effective only on models that are sensitive to massive weights, which could limit the method's practical applicability.
> > - The detection of massive activations and weights for a single model could yield different results with different datasets. The authors have not provided further analysis on how the choice of dataset affects the detection of these massive activations and weights, which might affect the reliability of the results.
> >
> > In view of above limitations, I plan to maintain my initial score.

---

> ### Author Response · Authors · 2024-12-03
> **Authors' opinions on the final concerns of the Reviewer 6SXM**
>
> Dear Reviewer 6SXM,
>
> We sincerely appreciate the time and effort you have devoted to providing thoughtful consideration and insightful feedback. We acknowledge and partially agree with the concerns you have raised. Nevertheless, we would like to provide **our perspective on the final concerns (C)**.
>
> ### **[C1&C2] Effectiveness of MacDrop and performance improvements achieved through MacDrop**
> - In our first manuscript, we demonstrated that some of LLMs (e.g., gemma-2 family) may not be sensitive to "massive" (**Figure 4**), and noted the potential limitations of our MacDrop (**Lines 334-335**).
>   - Reviewer 6SXM pointed it out, and we added the experiment results of these LLMs during the rebuttal (**Appendix F**). The results show that MacDrop is not effective for models that are not sensitive to massive weights.
>   - Thus, we agree that there are inherent limitations in MacDrop's practical applicability for the LLMs, which do not exhibit "massive" phenomena.
> - Nevertheless, we believe that **clearly demonstrating such limitations is itself meaningful**.
>   - Most previous and ongoing research addressing "massive" have stated that various models exhibit "massive" phenomena, and have proposed algorithms and experimental results excluding gemma-2 family, which does not exhibit "massive" phenomena.
>     - For instance, in the case of the streamingLLM algorithm [1], further verification may be required to determine whether retaining the KV of the initial tokens is truly effective for the gemma-2 family.
>   - Because we conducted **experiments on a wide variety of recent LLMs**, we were able to reveal such limitations.
>   - Additionally, we would like to emphasize that, similar to previous studies, MacDrop demonstrates sufficient feasibility for LLMs exhibiting "massive" phenomena.
> - About the improvement of MacDrop over LoRA and DoRA,
>   - First, we would like to clarify that MacDrop is not an algorithm designed to overcome the limitations of LoRA or DoRA, but an algorithm **applied orthogonally**. Rather, DoRA is the algorithm designed to overcome the limitations of LoRA.
>     - In **Table 2**, it can be observed that DoRA does not bring significant improvements compared to LoRA, in the zero-shot downstream tasks we evaluated.
>   - We agree with the opinion that the degree of performance improvement achieved through MacDrop is small.
>   - Nevertheless, it can be observed that **applying MacDrop to LoRA/DoRA results in greater performance improvement compared to the performance improvement of DoRA over LoRA**.
>   - Furthermore, during rebuttal, we provided the **robustness** under attacks (**Appendix G**) and performance on **long contexts** (**Appendix H**) of MacDrop. In these evaluations, MacDrop demonstrates a clear improvement.
>
>
> ### **[C3]  Impact of dataset selection on massive activations/weights**
> - We fully agree that understanding the relationship between datasets and massive activations/weights is important.
>   - For example, when Llama-3-8B is **pre-trained on dataset A**, massive activations appear; however, when Llama-3-8B is **pre-trained on dataset B**, massive activations may not appear.
> - Nonetheless, we would like to emphasize that this topic realistically requires **substantial resources** and should be treated as a standalone research.
>   - For instance, TinyLlama [2] requires **90 days using 16 A100-40G GPUs, for pre-training 1.1B Llama model on 3T tokens**.
>     - To understand the relationship with the dataset, various factors including data quantity and data quality must be considered. This would demand several times the resources mentioned above.
>     - Even with sufficient resources, it seems challenging to derive a meaningful answer within the rebuttal period.
>   - Furthemore, even if we were to observe the phenomenon at 1.1B, **raising the scaling issue (e.g., at 70B and 405B)** would still leave us unable to provide a definitive answer.
> - Therefore, to indirectly gain insights into this, we conducted research by using various open-source LLMs. (related to **W3.1** in our first response)
>   - Although the Llama-2/3/3.1 families are not a single model, they share very similar structures. And, we know that they were trained on different pre-training datasets (e.g., different cutoff dates).
>   - However, since all of these families exhibit "massive" phenomena, we inferred that the pre-training dataset itself does not significantly influence the manifestation of massive activations/weights.
>   - Based on this reasoning, we aimed to identify the other factors influencing the manifestation of massive activations/weights **from the architecture perspective** by comparing between Llama family and gemma-2 family, acknowledging that the pre-training dataset used for gemma-2 differs from those used for Llama. (**Lines 316-333**)
>
> Thank you.
> Authors.
>
> ---
> [1] Efficient Streaming Language Models with Attention Sinks, ICLR 2024
> [2] TinyLlama: An Open-Source Small Language Model, Arxiv 2024

---

### Comment · Area_Chair_45mk · 2024-11-28
**Reviewers, please kindly respond**

Dear Reviewers,

If you have not responded to author's rebuttal, please kindly do so as soon as possible. The deadline is Dec 2, but the authors can potentially further clarify questions if you respond earlier. Thanks!

Best, AC

---

> ### Author Response · Authors · 2024-11-28
>
> Dear AC,
>
> Thank you for your kind reminder to the reviewers to check the authors' responses. We truly appreciate your efforts in ensuring a thorough and fair review process.
>
> Dear reviewers,
>
> Please don’t hesitate to let us know if there’s any additional information or clarification we can provide.
>
> Best regards,
> Thank you.

---

### Meta-Review · Area_Chair_45mk · 2024-12-23

**Metareview:**

(a) Summary

The paper discovers massive weights as causes for massive activations, and proposes MacDrop, a dropout-based method targeting massive weights in LLMs, to improve fine-tuning efficiency and performance.

(b) Strengths

Identifies massive weights as causes for recent observations in LLMs; novel dropout-based fine-tuning for addressing reliance on massive activations; improvement in zero-shot downstream tasks

(c) Weaknesses

Limited experiments with larger models and diverse settings; marginal improvements in generation tasks; increased complexity from dropout curriculum (hence potential limited usage)

(d) Reasons for Decision

The limited experiments on datasets and models, and marginal gains, coupled with added complexity, do not outweigh the contributions. Not enough support from reviewers in general.

**Additional Comments On Reviewer Discussion:**

Overlap/novelty issues with methods like Wanda-SP and LoRA; authors highlighted focus on single-layer massive weights, and orthogonal usage with LoRA.

Limited applicability to models not sensitive to massive weights; authors demonstrated improvements on long-context tasks.

Discrepancy between perplexity and zero-shot scores; mostly clarified

More datasets and models; no new experiments are provided.

---

> ### Public Comment · ~Jaehoon_Oh1 · 2025-02-06
> **Authors' comments**
>
> We appreciate your thorough meta-review. However, we have noticed **potential inaccuracies** being discussed in this public space (OpenReview). To address these concerns, we would like to provide a response to clarify and correct the information.
>
>
> - About (a) Summary,
>   - The topic of **fine-tuning efficiency**, particularly in the context of speedup or compression, was not addressed in our paper.
>   - It seems that this expression was used because of parameter-efficient fine-tuning (PEFT). We are not proposing PEFT itself but rather a method that can be applied orthogonally to PEFT.
>
> - About (c) Weaknesses,
>   - ACs pointed out **limited experiments** with larger models and diverse settings.
>     - However, we analyzed **12 LLMs**, including Llama-3-70B and Llama-3.1-405B, and evaluated our method using most of these LLMs.
>     - We also provided the results according to **diverse dropout probabilities and scheduling**.
>     - About datasets, please refer to our reply "[C3] of Authors' opinions on the final concerns of the Reviewer 6SXM (Reviewer 6SXM)".
>   - ACs pointed out **increased complexity** from dropout curriculum.
>     - During the rebuttal with Reviewer ChSN, we checked MacDrop's overhead and answered that for Llama-3-8B using LoRA, there is approximately an additional computational cost of **0.35 seconds per step**, which is neglectable.
>     - In detail, please refer to our reply "[Q1] of Official Comment by Authors (Reviewer ChSN)".
> - About Additional Comments On Reviewer Discussion,
>   - The relationship between the question "Limited applicability to models not sensitive to massive weights" and the answer "Authors demonstrated improvements on long-context tasks." is incorrect.
>   - The proper answer is that "Authors **clearly showed limitations** of MacDrop by various experiments on a wide variety of recent LLMs, especially gemma-2 family, which is not senstive to massive weights. In fact, most previous studies have not examined this model."
>   - In detail, please refer to our reply "[C1&C2] of Authors' opinions on the final concerns of the Reviewer 6SXM (Reviewer 6SXM)".
>
>
> Best, Authors

---

### Decision · Program_Chairs · 2025-01-22

Reject